# The Development and Validation of a Novel “Dual Cocktail” Probe for Cytochrome P450s and Transporter Functions to Evaluate Pharmacokinetic Drug-Drug and Herb-Drug Interactions

**DOI:** 10.3390/pharmaceutics12100938

**Published:** 2020-09-30

**Authors:** Mihwa Kwon, Ji-Hyeon Jeon, Min-Koo Choi, Im-Sook Song

**Affiliations:** 1College of Pharmacy and Research Institute of Pharmaceutical Sciences, Kyungpook National University, Daegu 41566, Korea; mihwa_k@naver.com (M.K.); kei7016@naver.com (J.-H.J.); 2College of Pharmacy, Dankook University, Cheon-an 31116, Korea; minkoochoi@dankook.ac.kr; 3Vessel-Organ Interaction Research Center (VOICE), Kyungpook National University, Daegu 41566, Korea

**Keywords:** dual cocktail, pharmacokinetic drug-drug interaction, cytochrome P450 (CYP), transporter

## Abstract

This study was designed to develop and validate a 10 probe drug cocktail named “Dual Cocktail”, composed of caffeine (Cyp1a2 in rat and CYP1A2 in human, 1 mg/kg), diclofenac (Cyp2c11 in rat and CYP2C9 in human, 2 mg/kg), omeprazole (Cyp2c11 in rat and CYP2C19 in human, 2 mg/kg), dextromethorphan (Cyp2d2 in rat and CYP2D6 in human, 10 mg/kg), nifedipine (Cyp3a1 in rat and CYP3A4 in human, 0.5 mg/kg), metformin (Oct1/2 in rat and OCT1/2 in human, 0.5 mg/kg), furosemide (Oat1/3 in rat and OAT1/3 in human, 0.1 mg/kg), valsartan (Oatp2 in rat and OATP1B1/1B3 in human, 0.2 mg/kg), digoxin (P-gp in rat and human, 2 mg/kg), and methotrexate (Mrp2 in rat and MRP2 in human, 0.5 mg/kg), for the evaluation of pharmacokinetic drug–drug and herb-drug interactions through the modulation of a representative panel of CYP enzymes or transporters in rats. To ensure no interaction among the ten probe substrates, we developed a 2-step evaluation protocol. In the first step, the pharmacokinetic properties of five individual CYP probe substrates and five individual transporter substrates were compared with the pharmacokinetics of five CYP cocktail or five transporters cocktails in two groups of randomly assigned rats. Next, a pharmacokinetic comparison was conducted between the CYP or transporter cocktail group and the dual cocktail group, respectively. None of the ten comparison groups was found to be statistically significant, indicating the CYP and transporter substrate sets or dual cocktail set could be concomitantly administered in rats. The “Dual Cocktail” was further validated by assessing the metabolism of nifedipine and omeprazole, which was significantly reduced by a single oral dose of ketoconazole (10 mg/kg); however, no changes were observed in the pharmacokinetic parameters of other probe substrates. Additionally, multiple oral doses of rifampin (20 mg/kg) reduced the plasma concentrations of nifedipine and digoxin, although not any of the other substrates. In conclusion, the dual cocktail can be used to characterize potential pharmacokinetic drug–drug interactions by simultaneously monitoring the activity of multiple CYP isoforms and transporters.

## 1. Introduction

The evaluation of potential drug–drug interactions (DDIs) is becoming increasingly significant because of frequent concomitant use of multi-drugs among individuals over 60 years old, and the overuse of health supplements or over the counter drugs [1]. Owing to the growing use of herbal supplements, adverse drug reactions or herb–drug interactions (HDIs) resulting from the co-administration of therapeutic drugs with herbal supplements have also rapidly increased [2]. In China, herbal medicine formulations have accounted for approximately 13.8% and 17.3% of the total adverse drug reactions in 2010 and 2013, respectively [2]. The quantitative prediction of DDIs between co-administered therapeutic drugs using physiologically based pharmacokinetic modeling (PBPK) approaches has been widely used. However, PBPK approaches for HDIs between therapeutic drugs and herbal supplements have been limited to several herbs, such as St. John’s wort and milk thistle [3,4]. These approaches were performed using hyperforin and silybin, the major and well-characterized component of St. John’s wort and silymarin, respectively [3,4]. This limitation depends on the features of herbal medicines. Herbal supplements are bioactive constituent mixtures that substantially vary depending on the preparation methods [4]. Moreover, several bioactive constituents in one herbal supplement may coordinately interact with therapeutic drugs due to their similar inhibitory characteristics, although the pharmacokinetic features of these bioactive components have not been fully investigated [1,4]. In certain circumstances, such as when multiple components simultaneously inhibited the drug-metabolizing enzymes and transporters as well as when the pharmacokinetics of these components are not fully characterized, an in vivo cocktail approach may have advantages to efficiently evaluate potential DDIs or HDIs. The in vivo cocktail approach is a combination of several probe drugs that are sensitive to the modulation of specific metabolic enzymes or transporter proteins. This strategy is effective in characterizing the potential in vivo pharmacokinetic interactions between the probe drug and the potential perpetrator. Recent studies have focused on phenotyping major metabolizing enzymes in human and rat [5,6,7,8]. Among these cocktails “Inje cocktail [7]” which includes caffeine (cytochrome P450 [CYP] 1A2), omeprazole (CYP2C19), losartan (CYP2C9), dextromethorphan (CYP2D6), and midazolam (CYP3A), has been successfully used in rat cocktail studies [8]. In this study, significant DDIs in probe substrates were not detected even at high doses of caffeine (1 mg/kg), omeprazole (40 mg/kg), losartan (10 mg/kg), dextromethorphan (10 mg/kg), and midazolam (10 mg/kg) [8]. Videau et al. developed another cocktail set consisting of CYP enzyme and drug transporter modulators in both human and rat [9,10]. This cocktail set is composed of caffeine (CYP1A2), repaglinide (CYP2C8), tolbutamide (CYP2C9), omeprazole (CYP2C19), dextromethorphan (CYP2D6), midazolam (CYP3A), rosuvastatin (organic anion transporting polypeptides [OATPs]), acetaminophen (UGT), memantine (renal excretion), and digoxin (P-glycoprotein [P-gp]). The probe dose in rats is relatively low and within the range of 1–2 mg/kg [10]. Although this cocktail set included probes that are substrates for the OATPs and P-gp transporters, the use of transporter-targeted cocktails is still limited [11]. However, drug transporters are significant determinant of the pharmacokinetic features of drugs, and several clinically relevant DDIs or HDIs were mediated through drug transporter inhibition, as evidenced by numerous reports in the literature [1,12,13]. Therefore, this study aimed to develop a combination of dual cocktails that consists of five drugs as probes for CYP metabolizing enzyme function and five drugs as probes for transporter function to simultaneously monitor potential pharmacokinetic DDIs or HDIs in rats in vivo.

The probes were selected based on their safety profiles, ease of use, specificity, and the significance of their modifying metabolizing enzymes and transporters in pharmacokinetic DDIs according to the Food and Drug Administration (FDA) and European Medicines Agency (EMA) guidelines [14,15]. Notably, the lack of interaction among the cocktail substrates should have been demonstrated in vivo [15]. CYP1A2 (Cyp1a2 in rat), CYP2C9 (Cyp2c11 in rat), CYP2D6 (Cyp2d2 in rat), CYP2C19 (Cyp2c11 in rat), and CYP3A4 (Cyp3a1 in rat) were selected as the key drug metabolizing enzymes involved in DDIs and HDIs [7,8,16]. Caffeine, omeprazole, and dextromethorphan were selected as substrates for Cyp1a2, Cyp2c11, and Cyp2d2 based on the previous reports [7,8]. Nifedipine was selected as a substrate for Cyp3a1 based on previous results showing that the formation of dehydronifedipine from nifedipine is catalyzed by Cyp3a1 in rats, and nifedipine pharmacokinetics was modulated by the treatment of Cyp3a1 inhibitors [17,18]. Diclofenac was selected because Cyp2c11 is mainly involved in its metabolism [19]. The probes were used at the lowest dose that can reduce a DDI possibility among cocktail probe substrates according to the previously suggested in vivo animal cocktail (dose range, 0.5–2 mg/kg) [10]. In the case of dextromethorphan, we selected its dose as 10 mg/kg to access its plasma profile based on the study of Uchida et al. [8]. Therefore, to develop CYP cocktail set, caffeine (1 mg/kg), diclofenac (2 mg/kg), dextromethorphan (10 mg/kg), omeprazole (2 mg/kg), and nifedipine (0.5 mg/kg) were selected as their probe substrates [16,17,19].

P-gp (P-gp in rat), organic anion transporters (OATs; Oats in rat), organic cation transporters (OCTs; Octs in rat), multidrug resistance-associated protein 2 (MRP2; Mrp2 in rat), and OATPs (Oatps in rat) were selected as the key drug transporters involved in DDIs [11]. Digoxin (2 mg/kg), furosemide (0.1 mg/kg), metformin (0.5 mg/kg), methotrexate (0.5 mg/kg), and valsartan (0.2 mg/kg) were selected as their probe substrates because Octs, Oats, P-gp, and Mrp2 play significant roles in the disposition and plasma profile of these substrate drugs [11,13,20,21]. These substrates were intravenously administered due to their low bioavailability, which is below 30% [13,22,23,24,25,26]. The probe substrate doses were also selected as the lowest dose that could access full plasma profile in rat from the preliminary study. These substrates, which do not include any controlled substance, are typically used in both human and animals with optimal ease of access.

Here, we designed a two-step validation process to develop and validate the dual cocktail (Figure 1). In the first development phase, a cocktail of five CYP substrates and a cocktail of five transporter substrates combinations were separately validated to ensure no interactions exist among the selected substrates. In the second development phase, the cocktails of CYP substrates and transporter substrates were concurrently tested in rats to ensure that there are no interactions among these 10 probes. This dual cocktail was further validated with representative CYP or transporter inhibitors or inducers as well as red ginseng extract (RGE), a frequently used herbal supplement [27,28], to investigate the potential pharmacokinetic DDIs or HDIs associated with representative inhibitors or herbal supplements in rats.

RGE is one of the most popular herbal medicines in several countries, including East Asia [1]. Discrepancies have been observed in the HDIs of RGE in both human and animals. Gurley et al. [29,30] reported no significant CYP-mediated HDIs in young and elderly healthy subjects from the measurement of CYP1A2, CYP2E1, CYP3A4, and CYP2D6 activities following the oral administration of 0.5 g Korean ginseng powder twice daily (75 mg ginsenoside/day) for 28 days. In vivo cocktail approaches using a modified Inje cocktail to investigate HDIs between RGE (64% dried ginseng, once daily for two weeks, 85–100 mg ginsenoside/day) and CYP probe substrates in healthy Koreans reported a weak interaction of RGE with CYP2C9, CYP2C19, CYP3A4, and CYP2D6, but no interaction with P-gp and OATP1B1 [31,32]. In a study conducted by Malati et al. [33], Korean ginseng (0.5 g capsule twice daily for 28 days, 50 mg ginsenoside/day) induced CYP3A activity following the oral administration of 8 mg midazolam. In animal study, a repeated oral administration of RGE (0.5 g/kg for 2 weeks; equivalent to 4.05 mg ginsenoside/kg) did not alter the metabolic activity of Cyp1a, Cyp2b, Cyp2c, Cyp2d, and Cyp3a in the mouse liver [34]. However, another study revealed that the oral administration of the ethanol extract of ginseng (30 mg/kg for 10 days; equivalent to 8.13 mg ginsenoside/kg) to rats increased the mRNA expression of Cyp2d2 and Cyp3a1 in the rat liver [35]. Although the mechanism was not unveiled, the discrepancy between the two studies can be explained from different ginsenoside concentrations and compositions of the ginseng product used. For the conclusive investigation on the pharmacokinetic HDI of RGE, we selected a high-RGE dose (1.4 g/kg/day for 1 week, 26.5 mg ginsenosides/day) based on its therapeutic efficacy and investigated the modulatory effect of RGE on the 10 probe substrates for drug-metabolizing enzymes and transporters simultaneously using the developed “Dual Cocktail” set. RGE has been reported to be effective in various disease animal models, including diabetes, cancer, and arthritis, in the dose range of 50–2.0 g/kg (containing 3–25 mg/kg of total ginsenosides or 20 mg/kg ginsenoside Rc) [36,37,38,39,40,41].

## 2. Materials and Methods

### 2.1. Materials

Caffeine, diclofenac, dextromethorphan, nifedipine, omeprazole, digoxin, furosemide, metformin, methotrexate, valsartan, and berberine were purchased from Sigma-Aldrich (St. Louis, MO, USA). Paraxanthine, 4-hydroxy diclofenac, dextrorphan, dehydronifedipine, and 5-hydroxy omeprazole were purchased from Toronto Research Chemicals Inc. (North York, ON, Canada). All other chemicals and solvents are of reagent and HPLC grades.

RGE was obtained from Punggi Ginseng Cooperative Association (Punggi, Korea). RGE contains >60% of dried ginseng and total ginsenosides of 18.9 mg/g. This product was produced in the facilities under the current guidelines of the Korea Good Manufacturing Practice (Lot No. 71921092). Individual ginsenoside content is provided in Table 1.

### 2.2. Animals and Ethical Approval

Male Sprague-Dawley rats aged 7–8 weeks (Samtako, Osan, Korea) were housed under a 12 h light/dark cycle in a room with controlled temperature (24 ± 2 °C) and humidity (55 ± 5%). Animals were acclimatized for 1 week in an animal facility at Kyungpook National University. Food and water were provided ad libitum. All animal experiments were approved by Institutional Animal Care and Use Committee of Kyungpook National University (Approval No. KNU 2017-21, 17 February 2017).

### 2.3. Pharmacokinetic Study

For the development of 5 CYP cocktail (Figure 1), rats were randomly divided into the 5 CYP cocktail group (*n* = 6) and single groups (*n* = 6 each for individual probe substrate administration). The femoral arteries and femoral veins of rats were cannulated with PE50 polyethylene tubing (Jungdo, Seoul, Korea) under anesthesia with zoletil and lompun (50 and 5 mg/kg, respectively, intramuscular injection) and heparinized saline (10 U/mL) was used to prevent blood clotting. Pharmacokinetic studies were initiated after the recovery from anesthesia. Each rat in the 5 CYP cocktail group received the 5 CYP cocktail solution (2 mL/kg), including caffeine (1 mg/kg), diclofenac (2 mg/kg), dextromethorphan (10 mg/kg), omeprazole (2 mg/kg), and nifedipine (0.5 mg/kg). The probes were dissolved in a saline solution containing 10% DMSO. The rats in the single groups received individual CYP probe substrate solution orally with the same dose and volume as the cocktail solution. Blood samples were collected via the femoral artery at 0, 0.25, 0.5, 1, 2, 4, 8, and 24 h following the oral probe substrate administration, and a normal saline solution was administered via the femoral vein to compensate for blood sampling. After the centrifugation of blood samples at 8000× *g* for 1 min, 50 μL aliquots of plasma samples were stored at −80 °C until the analysis of probe substrates and their metabolites.

For the 5 transporter cocktail development (Figure 1), the rats were randomly divided into the 5 transporter cocktail (*n* = 6) and single groups (*n* = 6 each) for individual probe substrate studies. The femoral arteries and veins of the rats were cannulated with PE50 polyethylene tubing (Jungdo, Seoul, Korea) under anesthesia with Zoletil and lompun (50 and 5 mg/kg, respectively, intramuscular injection), and heparinized saline (10 U/mL) was used to prevent blood clotting. Pharmacokinetic studies were initiated after recovery from the anesthesia. Each rat in the 5 transporter cocktail groups received an intravenous cocktail (1 mL/kg) of 5 solutions, which included digoxin (2 mg/kg), furosemide (0.1 mg/kg), metformin (0.5 mg/kg), methotrexate (0.5 mg/kg), and valsartan (0.2 mg/kg). The drugs were dissolved in saline solution containing 10% DMSO. The rats within the single group studies received individual substrate solutions intravenously with an equivalent dose and volume to the cocktail solution. Blood samples were collected via the femoral artery at 0, 0.25, 0.5, 1, 2, 4, 8, and 24 h following the intravenous injection of probe substrates. Normal saline solutions were administered via the femoral vein to compensate for blood sampling. After the centrifugation of blood samples at 8000× *g* for 1 min, 50 μL aliquots of plasma samples were stored at −80 °C until the analysis of probe substrates.

For the dual cocktail development (Figure 1), the rats were randomly divided into three groups to receive the 5 CYP cocktail (*n* = 6), 5 transporter cocktail (*n* = 6), and dual cocktail set (*n* = 7). The femoral arteries and veins of rats were cannulated with PE50 polyethylene tubing (Jungdo, Seoul, Korea) under anesthesia with Zoletil and lompun (50 and 5 mg/kg, respectively, intramuscular injection), and heparinized saline (10 U/mL) was used to prevent blood clotting. Pharmacokinetic studies were started after the recovery from the anesthesia. The rats in the 5 CYP cocktail group received the 5 CYP substrate mixture orally. The rats in the 5 transporter cocktail group received the 5 transporter substrate mixture intravenously as previously described. The rats in the dual cocktail group simultaneously received the oral 5 CYP cocktail solution and the intravenous 5 transporter cocktail solution at an equivalent dose and similar method previously described. Blood samples were collected via the femoral artery at 0, 0.25, 0.5, 1, 2, 4, 8, and 24 h following the probe substrate administration.

For the CYP3A inhibition (Figure 1), the rats were randomly divided into the control (*n* = 3) and single ketoconazole (10 mg/kg) groups (*n* = 4). The rats in the ketoconazole group received ketoconazole solution (10 mg/mL/kg) via oral gavage, while the control rats received vehicle (1 mL/kg) also via oral gavage. After 1 h of ketoconazole treatment, the dual cocktail mixture solution was administered as previously described. Blood samples were collected via the femoral artery at 0, 0.25, 0.5, 1, 2, 4, 8, and 24 h following the probe substrate administration. For the OATPs inhibition (Figure 1), the rats were randomly divided into the control (*n* = 3) and single rifampin (20 mg/kg) groups (*n* = 4). The rats in the rifampin group received rifampin solution (20 mg/mL/kg) via oral gavage, while the control rats received vehicle (1 mL/kg) also via oral gavage. After 1 h of rifampin treatment, the dual cocktail mixture was administered as previously described. Blood samples were collected via the femoral artery at 0, 0.25, 0.5, 1, 2, 4, 8, and 24 h following the probe substrate administration. For the induction study (Figure 1), the rats were randomly divided into the control (*n* = 4) and multiple rifampin (20 mg/kg/day for 5 days) groups (*n* = 4). The rats in the rifampin group received rifampin solution (20 mg/mL/kg) via oral gavage, while the control rats received vehicle (1 mL/kg for 5 days) also via oral gavage. After 24 h of the last rifampin treatment, the dual cocktail mixture was administered as previously described to avoid the direct inhibitory effect of rifampin on drug transporters [42]. Blood samples were collected via the femoral artery at 0, 0.25, 0.5, 1, 2, 4, 8, and 24 h following the oral probe substrate administration. After the centrifugation of blood samples at 8000× *g* for 1 min, 50 μL aliquots of plasma samples were stored at −80 °C until the analysis of probe substrates and their metabolites.

For the dual cocktail application (Figure 1), the rats were randomly divided into the control (*n* = 7) and multiple RGE treatment (1.4 g/kg for 7 days) groups (*n* = 7). The rats in the multiple RGE group received RGE suspension (1.4 g/mL/kg/day) for 7 days orally via oral gavage. The control group received water (1 mL/kg) for 7 days by oral gavage. After 1 h of the last RGE treatment, the rats received the oral 5 CYP cocktail solution and simultaneously received the intravenous 5 transporter cocktail mixture at the same dose and method previously described. Blood samples were collected via the femoral artery at 0, 0.25, 0.5, 1, 2, 4, 8, and 24 h following the probe substrate administration. After the centrifugation of blood samples at 8000× *g* for 1 min, 50 μL aliquots of plasma samples were stored at −80 °C until the analysis of probe substrates and their metabolites.

The samples were prepared using protein precipitation. The samples (50 μL) were then added to 200 μL of internal standard (IS) working solution (berberine 2 ng/mL in methanol). After vortex mixing of the sample mixture for 15 min and centrifugation at 16,000× *g* for 5 min, 5 μL of the supernatant was injected into the LC-MS/MS system.

### 2.4. LC-MS/MS Analysis of Probe Substrate

The concentrations of 15 probe substrates and metabolites for CYP isoforms and transporters were simultaneously analyzed using an Agilent 6470 Triple Quadrupole LC-MS/MS system (Agilent, Wilmington, DE, USA). The ionization mode and mass transition for Q1 to Q3 were selected based on the product ion scan results of authentic standards and previously published reports [13,43,44,45,46,47,48,49]. The selected probe substrates and metabolites for CYP isoforms and transporters and their mass conditions are summarized in Table 2, including the ionization mode, mass transition, and collision energy. All analytes, with the exception of furosemide, were analyzed in the positive ionization mode.

In addition, the separation of 15 analytes and IS was performed on a Polar RP column (2.0 × 150 mm, 5 μm; Phenomenex, Torrance, CA, USA) using a mobile phase consisting of water: acetonitrile (30:70 *v*/*v*) with 0.1% formic acid at a 0.2 mL/min flow rate. The retention time and the linear range of the standard curve mixture of all analytes are also shown in Table 2. 

The intra- and inter-day precision and accuracy were analyzed for the six replicates at three QC samples (5, 50, and 250 ng/mL) for all analytes. Short-term stability was assessed by placing QC samples at 25 °C for 4 h. The stability of three freeze-thaw cycles was analyzed by comparing the QC samples (5 and 250 ng/mL) that underwent three freeze-thaw cycles (from −80 °C to 25 °C for 4 h as one cycle) with those of the control group. Post-treatment stability was evaluated by placing the processed QC samples in the autosampler at 6 °C for 24 h. 

### 2.5. Data Analysis

The pharmacokinetic parameters were calculated based on the plasma concentration-time profile using a non-compartment analysis by WinNonlin (version 5.1; Pharsights, Cary, NC, USA).

The statistical significance was assessed by a non-parametric Mann-Whitney test using the Statistical Package for the Social Sciences (version 24.0; SPSS Inc., Chicago, IL, USA). Differences between groups were considered to be significant when the *p*-values were under 0.05.

## 3. Results

### 3.1. Simultaneous Determination of Selected Probe Substrate for CYP Enzymes and Transporters

For the dual cocktail set development in rats, we applied a simultaneous determination of 15 substrates and metabolites in the rat plasma using an LC-MS/MS system based on the product ion scan results of authentic standards and previously published reports [13,43,44,45,46,47,48,49]. Representative multiple reaction-monitoring chromatograms for the selected probe substrates and metabolites of CYP enzymes and substrates for the transporters (Figure 2) showed that all the 15 probe substrates and metabolites were well separated with no interfering peaks at their respective retention times, suggesting that the 15 probe substrates and metabolites were simultaneously quantified in the rat plasma samples following the concomitant administration of 10 probe substrates. The lower limit of quantitation of the analytes were determined as 1 ng/mL for the 14 probe substrates or metabolites except for digoxin and 5 ng/mL for digoxin based on signal-to-noise ratios of over 10.0 [50]. The calibration curves showed good linearity over the range of 1–1000 ng/mL for CYP substrates and 1–2000 ng/mL for metformin, furosemide, valsartan, and methotrexate. The calibration curves of digoxin were linear in the concentration range of 5–2000 ng/mL (Table 2).

The precision and accuracy of intra- and inter-day assays are assessed using three different QC samples (5, 50, 250 ng/mL of each probe substrate) and results are shown in Table 3. In all cases, the intra- and inter-day precision and accuracy of the 15 analytes, including 10 probe substrates and five metabolites, are within 15% in the three QC sample concentrations (5, 50, and 250 ng/mL).

All analytes in plasma were stable for up to 4 h at 25 °C within 15% of standard deviation (SD) (Table 4). No significant degradation occurred in the low- and high-concentration QC samples of all analytes from the three freeze-thaw cycle stability and 24 h post-treatment stability measurement (Table 4). The results suggested that the analytical method developed using the LC-MS/MS system was reliable, reproducible, and accurate for the simultaneous analysis of the 10 CYP or transporter probe substrates and five CYP metabolites in rat plasma and, therefore, could be used for the development, validation, and application of the dual cocktail in rats. 

### 3.2. Comparison of Five CYP Probe Substrate Simultaneously versus Individually Administered

In the first step of five CYP cocktail development, we compared the plasma concentrations and the pharmacokinetic parameters of five CYP probe substrates between the cocktail and single administration groups. Figure 3A shows the plasma concentrations of caffeine and its metabolite, paraxanthine, in the single and CYP cocktail groups, respectively. The C_max_ and AUC values of caffeine were similar in the single and CYP cocktail groups. The metabolic ratio (MR) calculated from the AUC of paraxanthine to the AUC of caffeine were similar in the single and CYP cocktail groups (Table 5), suggesting remote interactions in the pharmacokinetics and metabolic activity of caffeine and other probe substrates. Similarly, the plasma concentrations of diclofenac and 4-hydroxy diclofenac (Figure 3B), omeprazole and 5-hydroxyomeprazole (Figure 3C), nifedipine and dehydronifedipine (Figure 3D), and dextromethorphan and dextrorphan (Figure 3E) and their pharmacokinetic parameters (Table 5) were compared between the control and CYP cocktail groups. Table 3 reveals that the pharmacokinetic parameters of the probe substrates (i.e., diclofenac, omeprazole, nifedipine, and dextromethorphan) and their metabolites in the CYP cocktail group were not statistically different in the single administration group: *p* > 0.05 for all parameters according to the Mann-Whitney U test. Collectively, these results suggested that 5 CYP cocktail substrates could be concomitantly administered to rats for the drug metabolic phenotype evaluation. The CYP cocktail included caffeine (1 mg/kg), diclofenac (2 mg/kg), omeprazole (2 mg/kg), nifedipine (0.5 mg/kg), and dextromethorphan (10 mg/kg) and was orally administered.

### 3.3. The Comparison of Five Transporter Probe Substrates Administered Simultaneously versus Individually

Next, we compared the plasma concentrations and pharmacokinetic parameters of the five probe substrates for transporters between the cocktail and single administration groups. Owing to the low bioavailability of probe substrates (less than 30%), including digoxin, furosemide, metformin, valsartan, and methotrexate [13,22,23,24,25,26], probe substrates were intravenously administered and renal excretion was also measured since transporters, such as Octs, Oats, P-gp, and Mrp2 play significant roles in the renal excretion of their substrate drugs [11]. Figure 4A shows the plasma concentrations of metformin in the single and transporter cocktail groups, respectively. The plasma AUC values and renal excretion of metformin in the cocktail group were similar to the single administration group (Table 6), suggesting the remote interactions between cocktail and single administration in metformin pharmacokinetics. Similarly, the plasma concentrations of furosemide (Figure 4B), valsartan (Figure 4C), digoxin (Figure 4D), and methotrexate (Figure 4E) and their pharmacokinetic parameters (Table 6) were compared between the control and transporter cocktail groups. Table 4 reveals that the pharmacokinetic parameters of the probe substrates (i.e., furosemide, valsartan, digoxin, and methotrexate) in the transporter cocktail group were not statistically different in the single administration group: *p* > 0.05 for all parameters according to the Mann-Whitney U test. Collectively, these results suggest that the five transporter cocktail substrates could be concomitantly administered to rats for the evaluation of the transporter activity and substrate drug pharmacokinetics. The cocktail that included metformin (0.5 mg/kg), furosemide (0.1 mg/kg), valsartan (0.2 mg/kg), digoxin (2 mg/kg), and methotrexate (0.5 mg/kg) was intravenously administered.

### 3.4. The Comparison of Pharmacokinetics Parameters Following Administration of Dual Cocktail vs. Five Transporter or Five CYP Cocktail

Next, we compared the pharmacokinetic parameters of the 10 probe substrates following the concomitant administration of an oral dose of five CYP cocktail and intravenous injection of five transporter cocktail in rats that received either the five CYP cocktail set or five transporter cocktail set, which were shown to be free of any interactions between individual probes (Figure 3 and Figure 4).

No significant difference was observed in the major pharmacokinetic parameters of 10 probe substrates administered as a separate five-CYP cocktail or five-transporter cocktail and dual cocktail (Figure 5 and Table 7). The results suggested that the administration of 10 probe substrates as dual cocktail could be used to phenotype the in vivo CYP or transporter activity or to evaluate potential DDIs.

### 3.5. Dual Cocktail Validation by Evaluating the Effects of Known CYP or Transporter Modulators on the Pharmacokinetic Parameters of Probe Substrates

To validate the dual cocktail set, a single oral dose of ketoconazole (10 mg/kg) as a representative Cyp3a inhibitor [51], a single oral dose of rifampin (20 mg/kg) as a representative Oatp inhibitor [21], and multiple oral doses of rifampin (20 mg/kg/dy for five days) as a representative Cyp3a and P-gp inducer [52,53] were administered before the dual cocktail substrates. Rifampin and ketoconazole has been widely used to investigate the pharmacokinetic DDIs in human as well as in animals [54,55].

As shown in Table 8, a single oral dose of ketoconazole significantly increased the plasma concentrations of nifedipine and omeprazole and consequently decreased the MR of nifedipine and omeprazole. Other substrates were not significantly changed by ketoconazole co-administration. The single oral administration of rifampin selectively inhibited the valsartan disposition and, therefore, increased the plasma exposure of valsartan by 3.5-fold, although the AUC values of other probe substrates and MR and AUC values of metabolites in the dual cocktail were not changed by the single rifampin treatment (Table 8), suggesting the selectivity of Oatp inhibition by single rifampin treatment and the feasibility of the dual cocktail in assessing pharmacokinetic DDIs. Repeated rifampin administration, which is known for inducing Cyp3a and P-gp [52,53], increased the disposition of nifedipine and digoxin, which consequently decreased the AUC values of nifedipine and digoxin (Table 8). The AUC values of other substrate drugs were not altered by the repeated rifampin administration.

In summary, the dual cocktail of 10 probe substrates for five CYPs and five transporters was successfully developed without any significant pharmacokinetic DDIs among the probe substrates. This dual cocktail was validated using a representative Cyp3a inhibitor ketoconazole, an Oatp inhibitor rifampin (single administration) [21], and a repeated rifampin administration, which acts as a Cyp3a and P-gp inducer [52,53].

However, the AUC values of dehydronifedipine were not significantly changed by the repeated rifampin treatment, and the AUC values of the metabolites and MR of nifedipine might not indicate Cyp3a activity modulation. In omeprazole and dextromethorphan cases, the MR of these substrates could be compromised owing to multiple enzyme involvement in their metabolism [56,57]. Moreover, since the major elimination route of valsartan and methotrexate is biliary excretion [20,58,59], Xu could not represent the transport activity of Octs, Oatps, P-gp, and Mrp2. Therefore, the phenotypic index for the assessment of potential pharmacokinetic DDIs could be suggested as the changes in the AUC values of probe substrates (Table 9). 

### 3.6. The Use of the Dual Cocktail to Assess the Pharmacokinetic Herb-Drug Interaction of RGE Treatment

To further validate the utility of the dual cocktail, the rats were pretreated with RGE, and the dual cocktail was administered to assess the pharmacokinetic herb–drug interactions between probe substrates and RGE following the oral administration of RGE (1.4 g/kg/day) for seven days.

The plasma concentrations of probe substrates are shown in Figure 6. The plasma concentration profiles of caffeine, dichlofenac, omeprazole, dextromethorphan, valsartan, and digoxin in the RGE group were similar to those in the control group. However, the plasma concentrations of metformin, furosemide, and methotrexate in the RGE group were higher than those in the control group (Figure 6).

The comparison of AUC values of the probe substrates between the control and RGE groups showed no significant pharmacokinetic herb–drug interaction involving caffeine, dichlofenac, omeprazole, dextromethorphan, valsartan, and digoxin after the repeated RGE treatment. However, the AUC values of nifedipine, metformin, furosemide, and methotrexate following the RGE treatment were significantly greater than the control group (Figure 7). The results suggested that the repeated RGE administration decreased the activity of Cyp3a, Oct, Oat, and Mrp2, and subsequently decreased the disposition of their specific probe substrate drugs, including nifedipine, metformin, furosemide, and methotrexate, respectively. The observed pharmacokinetic interactions between RGE and metformin and methotrexate are consistent with previous results [12,21]. The lack of pharmacokinetic interaction between valsartan and RGE is also consistent with previous reports [21].

## 4. Discussion

With an increasing relevance of evaluating potential DDIs and HDIs, there is a need to increase the efficiency and speed of identifying these undesirable events. In the case of therapeutic drugs, in vitro to in vivo extrapolation and in silico PBPK modeling have been typically used for predicting DDIs [60]. Despite the increasing likelihood of HDIs, a standard in silico prediction system for evaluating HDIs remains elusive [61]. Currently, in vitro HDI evaluation in in vitro systems and preclinical animal study to estimate the inhibition and/or induction of drug-metabolizing enzymes and transporters are frequently used [61]. The separate investigation of potential CYP metabolizing enzymes and transporter mediated interactions can be a costly and time-consuming process. Therefore, several cocktail-based approaches to assess the potential in vitro and in vivo DDIs and HDIs have been developed [5,7,8,9,10]. Here, we developed a cocktail set composed of ten probe drugs for phenotyping both CYP metabolic enzymes and transporters in animal models simultaneously.

To develop the cocktail set, the simultaneous assessment of the probe substrates and their metabolites is critical. Here, we successfully developed the simultaneous determination method of 15 substrates and metabolites in the plasma with concentrations that ranged from 1 ng/mL (5 ng/mL for digoxin) to 1000 (for CYP substrates) or 2000 ng/mL (for transporter substrates). With the use of simultaneous analytical method of probe substrates and a series of well-designed pharmacokinetic studies, we demonstrated the lack of pharmacokinetic DDI among the 10 probe substrates (Table 5, Table 6 and Table 7).

The composition of the dual cocktail was determined according to the following probe drug selection criteria: (i) major CYP metabolizing enzymes and transporters in both human and rat; (ii) the safety and ease of access of probe drugs; (iii) the selectivity of probe substrates to the CYP enzymes or transporters of interest. The CYP cocktail set included probes for Cyp1a2, Cyp2c11, Cyp2d2, and Cyp3a1, which were selected based on previous cocktail sets developed for human and rat studies [7,8,9,10] because the consistency of the probe substrates of the in vivo cocktail among human and rats are critical in estimating the potential in vivo DDIs based on animal data. The homology of CYP orthologues between human and rats are approximately 70–75% for CYP1A2 (Cyp1a2), CYP2C9 (Cyp2c11), CYP2D6 (Cyp2d2), CYP3A4 (Cyp3a1), although largely unknown for CYP2C19 (Cyp2c11) [62,63,64]. Cocktails designed with high consistency have great values in phenotyping metabolic enzyme and transporter activities in rats as models for different disease states [8,10]. They can be used to assess potential pharmacokinetic HDIs for herbal supplements that might be concomitantly administered with therapeutic drugs. The five probes CYP substrates, which included caffeine (Cyp1a2 in rat and CYP1A2 in human, 1 mg/kg), diclofenac (Cyp2c11 in rat and CYP2C9 in human, 2 mg/kg), omeprazole (Cyp2c11 in rat and CYP2C19 in human, 2 mg/kg), dextromethorphan (Cyp2d2 in rat and CYP2D6 in human, 10 mg/kg), nifedipine (Cyp3a1 in rat and CYP3A4 in human, 0.5 mg/kg), were selected based on their use in other cocktails [7,18]. CYP2D6 is responsible for the metabolism of dextromethorphan to its major metabolite, dextrorphan. Dextromethorphan is also metabolized to 3-methoxymorphinan by CYP3A4 and subsequently metabolized to 3-hydroxymorphinan by CYP2D6. The formation of 3-methoxymorphinan by CYP3A4 is 2–17% of dextrorphan formation [57]. A more than 200-fold decrease in dextrorphan formation was observed by the CYP2D6 compared with CYP2D6 poor metabolizer, suggesting the major contribution of CYP2D6 in the dextromethorphan metabolism [57]. In our study, no significant alterations in the plasma concentration of dextromethorphan by the ketoconazole treatment indicated the minor contribution of Cyp3a in the dextromethorphan pharmacokinetics. However, the involvement of multiple enzymes such as Cyp3a and Cyp2d and subsequent metabolic pathway from dextrorphan to hydroxymorphinan could be a confounding factor in pharmacokinetic DDIs or HDIs. Similar case should also be considered in omeprazole metabolism: Cyp3a-mediated omeprazole sulfone formation and Cyp2c-mediated 5-hydroxyomeprazole formation [56].

Meanwhile, we developed a dual cocktail that combines both CYP and transporter substrates owing to limited studies of substrate cocktails for drug transporter studies. The transporter cocktail set included probes for Oct1/2, Oat1/3, Oatp2, P-gp, and Mrp2, which play significant roles in the distribution and elimination of substrate drugs in the livers and kidneys [11]. Furthermore, these transporters shared similar substrate specificity among different species [65,66]. For example, an evaluation of 3300 Pfizer compounds in human P-gp showed fewer differences in substrate susceptibility than rodent P-gp [67], and the functional orthologues of human OCTs, OATs, OATPs, P-gp, and MRPs were identified in mice and rats [65]. The transporter substrate cocktail set included metformin (Oct1/2 in rat and OCT1/2 in human, 0.5 mg/kg), furosemide (Oat1/3 in rat and OAT1/3 in human, 0.1 mg/kg), valsartan (Oatp2 in rat and OATP1B1/1B3 in human, 0.2 mg/kg), digoxin (P-gp in rat and human, 2 mg/kg), and methotrexate (Mrp2 in rat and MRP2 in human, 0.5 mg/kg) as probe substrates. These substrates were intravenously administered due to their low bioavailability, which is below 30% in rats [13,22,23,24,25,26]. Their renal excretion was assessed since transporters, such as Oct, Oat, P-gp, and Mrp play significant roles in their substrate drug excretion [11,13,20,21]. Therefore, the transporter cocktail set could be used for the determination of pharmacokinetic alteration and disposition of probe substrates by the co-administered drugs or herbal supplements. However, we should note that transporter mediated DDIs and HDIs occurring in the intestinal absorption could not be detected in our dual cocktail system because of the intravenous injection of transporter cocktail set. Therefore, if the perpetrator is likely to interact with intestinal transporters such as P-gp, oral administrations of probe substrate such as digoxin concomitantly with perpetrator should be considered to investigate the intestinal DDIs or HDIs.

In addition, the dual cocktail set is designed to determine the pharmacokinetic DDIs or HDI related to the five CYP and five transporter activity modulation in rats; therefore, the investigation of pharmacodynamic interactions that might occur among the probe substrates or between probe substrate and perpetrator drug or herbal supplement could not be identified by the single low-dose administration of dual cocktail probe substrates.

An increased renal toxicity and a decreased renal excretion of methotrexate by the co-medication of nonsteroidal anti-inflammatory drugs, such as diclofenac, in the treatment of rheumatoid arthritis has been reported [68]. As the underlying mechanism, diclofenac-mediated OAT3 inhibition has been proposed with the IC_50_ values of diclofenac of 6.13 μM [69]. However, Oat inhibition by diclofenac is unlikely in our cocktail set because the unbound maximum plasma concentration of diclofenac was calculated as 0.06 μM. The DDIs in hypertensive patients who were prescribed with antihypertensive, cardiovascular, and antidiabetic drugs are prevalent, and more than 55% and 32% of these DDIs were pharmacodynamic synergisms or antagonisms and increased serum potassium levels [70]. In our cocktail set, some antihypertensive drugs, such as nifedipine, furosemide, valsartan, and co-medicated antidiabetic and cardiovascular drugs, such as metformin and digoxin, could have possibility of these pharmacodynamic DDIs or altered potassium levels. However, as mentioned earlier, these pharmacodynamic DDIs or altered potassium levels is challenging to detect from the single administration of our dual cocktail set.

We also should note that large variations in the plasma concentrations of probe substrates, such as caffeine, omeprazole, nifedipine, dextromethorphan, metformin, and digoxin, at approximately 4–8 h that also resulted in the large variations in the AUC values of these substrates were observed during the development and validation stage of the dual cocktail set. Large variations in the urinal excretion (Xu) were also observed in metformin and digoxin. These results are attributed to the difference in the elimination rate of individual rats. That is, two or three out of six rats showed fast elimination compared with other rats in our study group (Table 5, Table 6 and Table 7).

In the validation stage, a single oral dose of rifampin increased the plasma concentration of valsartan, which is due to the OATP inhibition by rifampin, and is consistent with the previous reports [21]; however, the plasma concentrations of other substrates were not changed by the single oral rifampin administration (Table 6). Multiple rifampin administration is known for inducing PXR and, consequently, inducing the CYP3A4 and P-gp transcription [54,55,71]. To avoid the direct inhibition by rifampin, the dual cocktail was administered 24 h after the last rifampin administration, according to the previous reports [42]. As expected, multiple rifampin treatment decreased the plasma concentrations of nifedipine and digoxin, which is caused by the increased CYP3A4 and P-gp activity. A single oral dose of ketoconazole increased the plasma concentrations of nifedipine and omeprazole. Based on the inhibitory effect of ketoconazole on CYP3a [72], a decreased dehydronifedipine formation and an increased nifedipine concentration in rat plasma could be expected, which was consistent with previous reports [17,18]. In case of omeprazole, CYP3A and CYP2C19 are involved in the omeprazole metabolism resulting in the formation of omeprazole sulfone and 5-hydroxyomeprazole, respectively [56]. Therefore, the ketoconazole treatment might inhibit the CYP3A4-mediated omeprazole metabolism and increase the plasma concentration of omeprazole; however, could not alter the formation of 5-hydroxyomeprazole (Table 6).

The concomitant administration of omeprazole decreased the itraconazole plasma concentration by decreasing its pH-dependent dissolution in human [73], although did not significantly alter the plasma concentrations of itraconazole when orally administered [74]. It suggests the interaction between omeprazole and azole antifungal agent during the dissolution and absorption phase. In our study, we orally administered ketoconazole and administered omeprazole 1 h later at a dose of 2 mg/kg as the cocktail set. Considering that the T_max_ of ketoconazole is 0.5–1 h in rats [75] and the effective omeprazole dose to increase gastric pH in rat is 20 mg/kg [76], the interaction between ketoconazole, and omeprazole during the absorption phase could be disregarded in this study.

Ketoconazole inhibited P-gp and CYP3A4 in both human and rats [77,78]. In humans, the in vivo K_i_ value of ketoconazole for hepatic CYP3A4 and renal P-gp was estimated to be 6.64 and 2.27 ng/mL, respectively [77]. In rats, the oral administration of ketoconazole at a dose of 50 mg/kg significantly increased the plasma concentrations of TAK-427, a P-gp substrate drug, following the oral and intravenous administration of TAK-427 [78]. However, in our case, we orally administered ketoconazole at a dose of 10 mg/kg in rats, which did not cause a significant alteration in the pharmacokinetics of digoxin, a probe substrate for P-gp (Table 6). This difference could be explained by the different dose of ketoconazole (50 mg/kg in the study of Takeuchi et al. [78] and 10 mg/kg in this study). Collectively, these results demonstrate the selectivity of known inhibitors, such as ketoconazole and rifampin, and known inducer rifampin in the assessment of pharmacokinetic DDIs of the dual cocktail set in rats. In addition, we used the novel dual cocktail set to detect the pharmacokinetic herb–drug interactions of RGE, one of the most popular herbal supplements. RGE pretreatment increased the plasma concentration of metformin, which is a specific probe for the Oct transporter. We previously demonstrated the tissue-specific regulation of Oct1 transporters in the intestine, liver, and kidneys [12]. The increased Oct1 expression in the enterocytes and decreased hepatic Oct1 expression coordinately increased the plasma concentration of metformin [12]. In this study, since metformin was intravenously administered, the increased plasma concentration of metformin could be due to the decreased hepatic Oct1 function. Similarly, Shu et al. [79] reported that the decreased OCT1 expression and function increased the plasma concentration of metformin in human. We also previously demonstrated the decreased Mrp2 expression in the rat liver following the repeated RGE treatment (1.5 g/kg/day for seven days) and, consequently, the biliary excretion of methotrexate was decreased leading to an increased methotrexate plasma concentration [13], which is consistent with the observations in this study (Figure 7J). In addition, the plasma exposure of nifedipine and furosemide was also increased following the repeated RGE administration, which indicates the Cyp3a and OATS inhibition by RGE. The underlying mechanism(s) on the expression and function of Cyp3a and OATS by the multiple RGE exposure need to be further investigated. Taken together, the results demonstrate the applicability of the dual cocktail in effectively evaluating the pharmacokinetic herb–drug or drug–drug interactions by the simultaneous phenotyping of the activities of five CYP enzymes (Cyp1a, 2c, C9, 2C19, 2D6, and 3A4) and five drug transporters (OCTs, OATs, OATPs, P-gp, and MRP2).

In conclusion, we developed and validated a dual cocktail set, which is composed of 10 probe substrates, including caffeine (Cyp1a2 in rat and CYP1A2 in human, 1 mg/kg), diclofenac (Cyp2c11 in rat and CYP2C9 in human, 2 mg/kg), omeprazole (Cyp2c11 in rat and CYP2C19 in human, 2 mg/kg), dextromethorphan (Cyp2d2 in rat and CYP2D6 in human, 10 mg/kg), nifedipine (Cyp3a1 in rat and CYP3A4 in human, 0.5 mg/kg), metformin (Oct1/2 in rat and OCT1/2 in human, 0.5 mg/kg), furosemide (Oat1/3 in rat and OAT1/3 in human, 0.1 mg/kg), valsartan (Oatp2 in rat and OATP1B1/1B3 in human, 0.2 mg/kg), digoxin (P-gp in rat and human, 2 mg/kg), and methotrexate (Mrp2 in rat and MRP2 in human, 0.5 mg/kg). This dual cocktail set can be used to efficiently and simultaneously characterize the activity of five CYP enzymes and five transporters and effectively evaluate the pharmacokinetic drug–drug and herb–drug interactions through the modulation of five CYP enzymes and five transporters in rats.

## Figures and Tables

**Figure 1 pharmaceutics-12-00938-f001:**
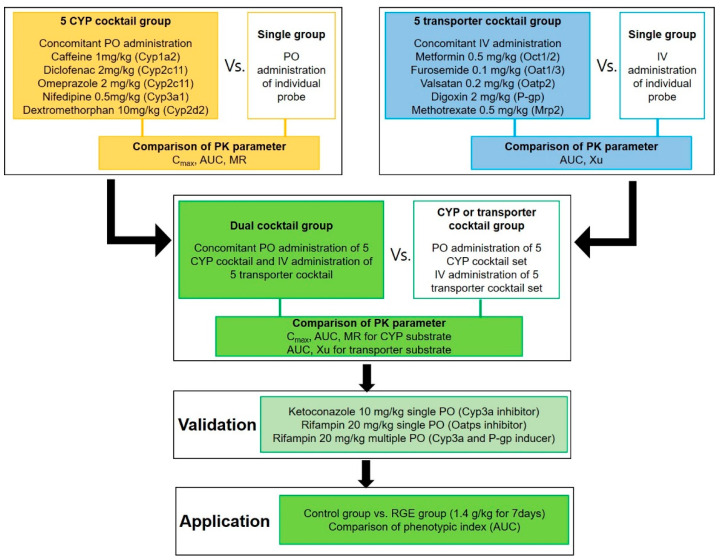
Strategy for the “Dual Cocktail” development. CYP; cytochrome P450, PO; per oral, IV; intravenous, PK, Pharmacokinetics, C_max_; maximum plasma concentration, AUC; area under plasma concentration curve, MR; metabolic ratio calculated by dividing metabolite AUC by parent AUC, Xu; excreted amount of substrate via renal route, RGE; red ginseng extract.

**Figure 2 pharmaceutics-12-00938-f002:**
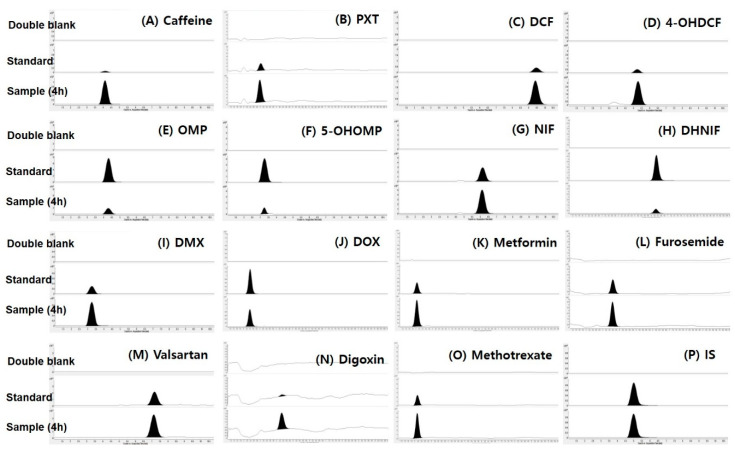
Representative multiple reaction-monitoring (MRM) chromatograms of the the selected probe substrates and metabolites for Cytochrome P450 (CYP) enzymes and substrates for transporters in rat double blank plasma (upper panel), rat blank plasma is spiked with a standard solution (5 ng/mL) of 15 probe substrates or metabolites to rats (middle panel), and rat plasma samples at 4 h following dual cocktail co-administration (lower panel). (**A**) Caffeine, (**B**) paraxanthine (PXT), (**C**) diclofenac (DCF), (**D**) 4-hydroxy diclofenac (4-OHDCF), (**E**) omeprazole (OMP), (**F**) 5-hydroxyomeprazole (5-OHOMP), (**G**) nifedipine (NIF), (**H**) dehydronifedipine (DHNIF), (**I**) dextromethorphan (DMX), (**J**) dextrorphan (DOX), (**K**) metformin, (**L**) furosemide, (**M**) valsartan, (**N**) digoxin, (**O**) methotrexate, and (**P**) berberine (IS).

**Figure 3 pharmaceutics-12-00938-f003:**
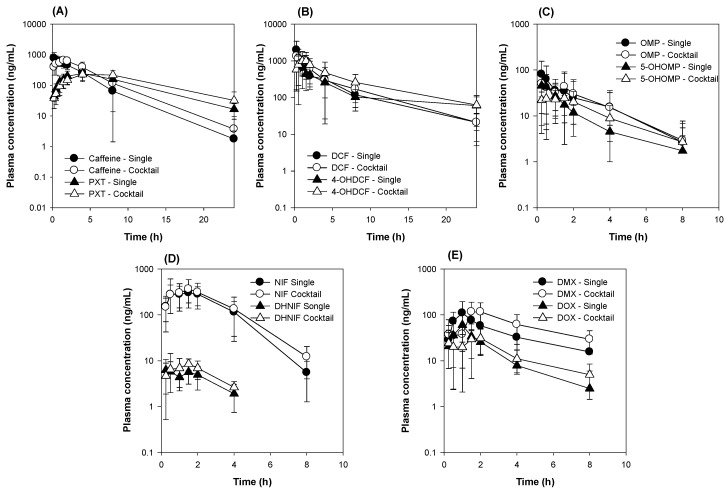
The plasma concentrations of (**A**) caffeine and paraxanthine (PXT), (**B**) diclofenac (DCF) and 4-hydroxy diclofenac (4-OHDCF), (**C**) omeprazole (OMP) and 5-hydroxyomeprazole (5-OHOMP), (**D**) nifedipine (NIF) and dehydronifedipine (DHNIF), and (**E**) dextromethorphan (DMX) and dextrorphan (DOX) following the single (closed symbol) and CYP cocktail (open symbol) oral administration of caffeine (1 mg/kg), diclofenac (2 mg/kg), omeprazole (2 mg/kg), nifedipine (0.5 mg/kg), and dextromethorphan (10 mg/kg) in rats. The data is expressed as the mean ± SD (*n* = 6).

**Figure 4 pharmaceutics-12-00938-f004:**
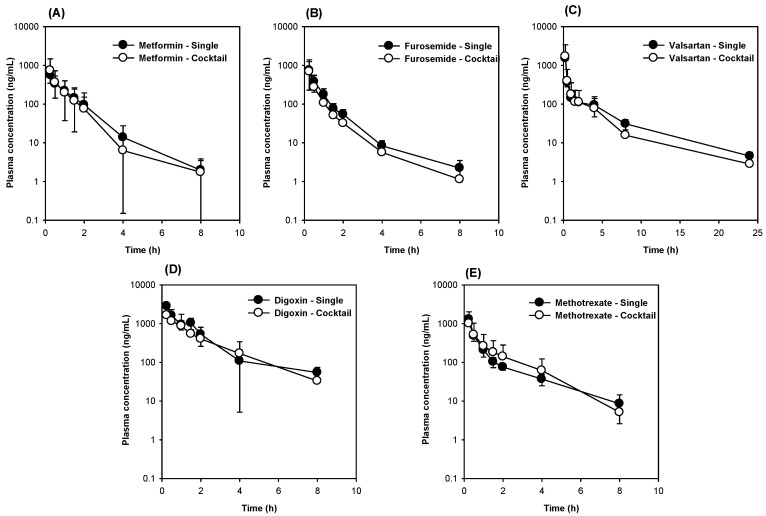
The plasma concentrations of (**A**) metformin, (**B**) furosemide, (**C**) valsartan, (**D**) digoxin, and (**E**) methotrexate following the single (closed symbol) and transporter cocktail (open symbol) administration of metformin (0.5 mg/kg), furosemide (0.1 mg/kg), valsartan (0.2 mg/kg), digoxin (2 mg/kg), and methotrexate (0.5 mg/kg) intravenous injection in rats. The data are presented as the means ± SD (*n* = 6).

**Figure 5 pharmaceutics-12-00938-f005:**
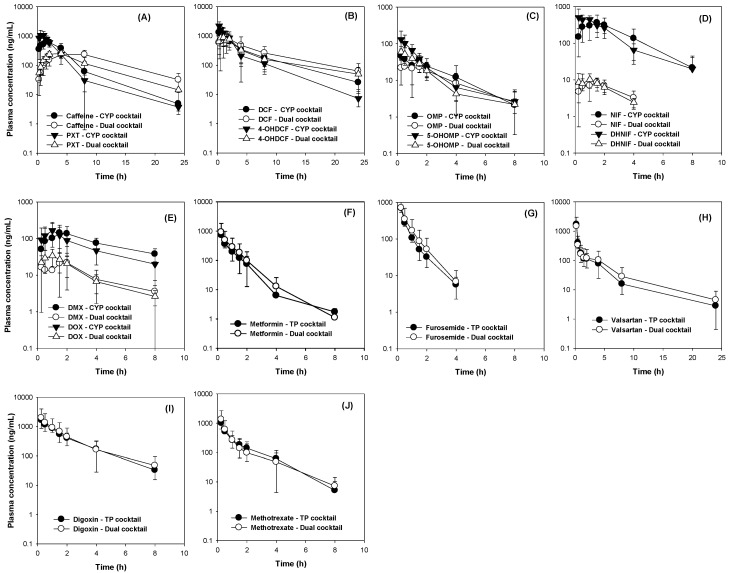
The plasma concentrations of (**A**) caffeine and paraxanthine (PXT), (**B**) diclofenac (DCF) and 4-hydroxy diclofenac (4-OHDCF), (**C**) omeprazole (OMP) and 5-hydroxyomeprazole (5-OHOMP), (**D**) nifedipine (NIF) and dehydronifedipine (DHNIF), and (**E**) dextromethorphan (DMX) and dextrorphan (DOX) following the administration of five CYP cocktail (closed symbol) and dual cocktail (open symbol) in rats (*n* = 6). The plasma concentrations of (**F**) metformin, (**G**) furosemide, (**H**) valsartan, (**I**) digoxin, and (**J**) methotrexate following the administration of transporter (TP) cocktail (closed symbol) and dual cocktail (open symbol) in rats (*n* = 6). The CYP cocktail consists of caffeine (1 mg/kg), diclofenac (2 mg/kg), omeprazole (2 mg/kg), nifedipine (0.5 mg/kg), and dextromethorphan (10 mg/kg) and orally administered. The TP cocktail consists of metformin (0.5 mg/kg), furosemide (0.1 mg/kg), valsartan (0.2 mg/kg), digoxin (2 mg/kg), and methotrexate (0.5 mg/kg) and administered via intravenous injection. Dual cocktail is a mixture of CYP and TP cocktails. The data are presented as the means ± SD (*n* = 6 for CYP cocktail or transporter cocktail, *n* = 7 for dual cocktail).

**Figure 6 pharmaceutics-12-00938-f006:**
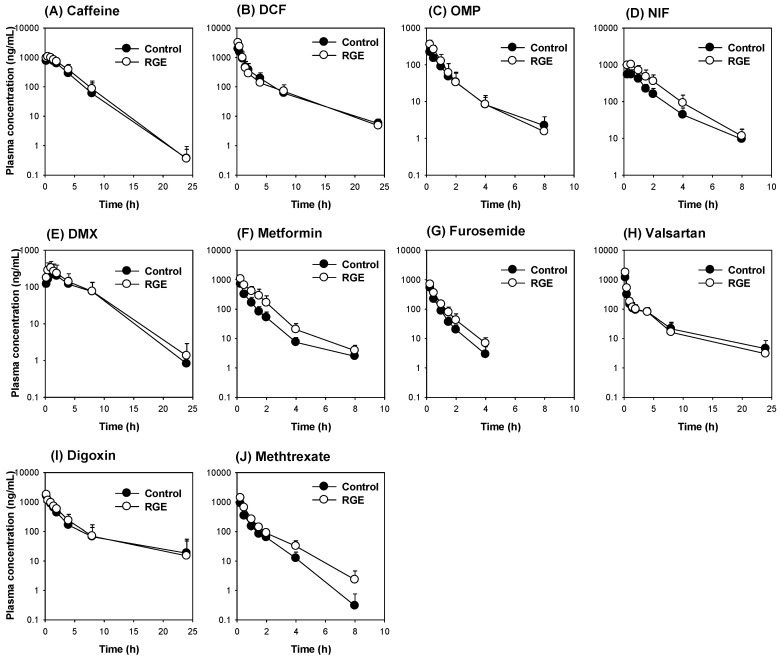
The plasma concentrations of (**A**) caffeine, (**B**) diclofenac (DCF), (**C**) omeprazole (OMP), (**D**) nifedipine (NIF), (**E**) dextromethorphan (DMX), (**F**) metformin, (**G**) furosemide, (**H**) valsartan, (**I**) digoxin, and (**J**) methotrexate following the repeated RGE administration (1.5 g/kg/day for 7 days) in rats that received dual cocktail sets. Dual cocktail set is composed of a mixture of CYP and transporter cocktails. The CYP cocktail consists of caffeine (1 mg/kg), diclofenac (2 mg/kg), omeprazole (2 mg/kg), nifedipine (0.5 mg/kg), and dextromethorphan (10 mg/kg) and orally administered. The transporter cocktail mixture is composed of metformin (0.5 mg/kg), furosemide (0.1 mg/kg), valsartan (0.2 mg/kg), digoxin (2 mg/kg), and methotrexate (0.5 mg/kg) and administered via intravenous injection. The data are presented as the means ± SD (*n* = 7).

**Figure 7 pharmaceutics-12-00938-f007:**
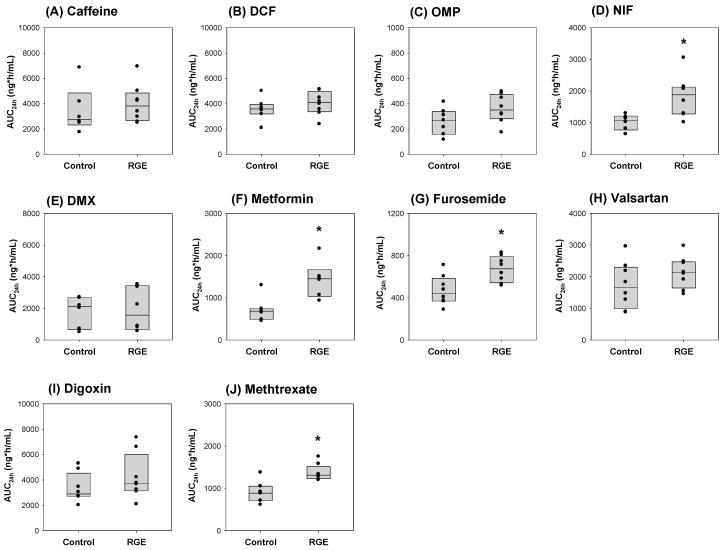
The comparison of AUC values of (**A**) caffeine, (**B**) diclofenac (DCF), (**C**) omeprazole (OMP), (**D**) nifedipine (NIF), (**E**) dextromethorphan (DMX), (**F**) metformin, (**G**) furosemide, (**H**) valsartan, (**I**) digoxin, and (**J**) methotrexate between the control and RGE treatment groups. * *p* < 0.05 is considered statistically significant compared with the control group by the Mann-Whitney U test. The data are presented as the means ± SD (*n* = 7).

**Table 1 pharmaceutics-12-00938-t001:** Ginsenoside content in red ginseng extract (RGE) product.

Ginsenosides		Content (mg/g RGE)	Total (mg/g RGE)
Protopanaxadiol-type	Ginsenoside Rb1	4.7	14.3
	Ginsenoside Rb2	2.3	
	Ginsenoside Rc	2.5	
	Ginsenoside Rd	1.3	
	Ginsenoside Rg3	3.5	
Protopanaxatriol-type	Ginsenoside Re	1.3	4.6
	Ginsenoside Rg1	0.6	
	Ginsenoside Rf	1.1	
	Ginsenoside Rh1	1.6	

**Table 2 pharmaceutics-12-00938-t002:** MS/MS parameters for the detection of 15 probe substrates and metabolites.

Analyte	Mode	*m*/*z* Q1 → Q3	Collision Energy (V)	Retention Time (min)	Linear Range (ng/mL)
Caffeine	positive	194.9 → 138.0	20	4.09	1–1000
Paraxanthine	positive	180.7 → 124.3	10	2.97	1–1000
Diclofenac	positive	295.9 → 215.0	20	9.58	1–1000
4-Hydroxydiclofenac	positive	312.2 → 230.9	20	5.25	1–1000
Dextromethorphan	positive	272.0 → 214.9	25	3.37	1–1000
Dextrorphan	positive	258.0 → 201.0	25	2.42	1–1000
Nifedipine	positive	347.0 → 315.0	5	6.16	1–1000
Dehydronifedipine	positive	245.0 → 283.9	30	6.44	1–1000
Omeprazole	positive	346.0 → 197.8	5	4.29	1–1000
5-Hydroxyomeprazole	positive	362.2 → 213.9	10	3.26	1–1000
Digoxin	positive	803.4 → 283.0	50	4.35	5–2000
Furosemide	negative	328.9 → 284.8	15	3.62	1–2000
Metformin	positive	129.9 → 60.0	10	2.05	1–2000
Methotrexate	positive	455.1 → 308.2	20	2.23	1–2000
Valsartan	positive	436.1 → 291.0	5	7.22	1–2000
Berberine	positive	336.1 → 320.0	30	5.07	

**Table 3 pharmaceutics-12-00938-t003:** Intra- and inter-day precision and accuracy of 15 probe substrates and metabolites.

Analyte	Spiked Concentration (ng/mL)	Intra-Day			Inter-Day		
		Measured (ng/mL)	Precision (%)	Accuracy (%)	Measured (ng/mL)	Precision (%)	Accuracy (%)
Caffeine	5	4.65 ± 0.41	8.78	93.07	4.78 ± 0.41	8.66	95.51
	50	46.42 ± 1.94	4.18	92.84	48.31 ± 5.70	11.81	96.63
	250	253.58 ± 11.79	4.65	101.43	273.24 ± 24.11	8.83	109.30
Paraxanthine	5	5.17 ± 0.40	7.76	103.44	5.17 ± 0.40	7.76	103.44
	50	43.66 ± 1.89	4.33	87.32	50.59 ± 6.72	13.28	101.18
	250	222.27 ± 2.77	1.25	88.91	270.32 ± 33.37	12.34	108.13
Diclofenac	5	5.05 ± 0.42	8.35	100.90	5.17 ± 0.55	10.73	103.34
	50	47.90 ± 0.38	0.79	95.80	47.89 ± 5.28	11.02	95.77
	250	248.34 ± 20.03	8.07	99.34	244.29 ± 16.02	6.56	97.72
4-Hydroxy diclofenac	5	4.54 ± 0.31	6.91	90.76	4.68 ± 0.43	9.24	93.51
	50	49.19 ± 1.59	3.23	98.38	47.95 ± 5.19	10.83	95.90
	250	261.64 ± 18.42	7.04	104.66	271.16 ± 24.18	8.92	108.47
Omeprazole	5	4.46 ± 0.07	1.65	89.27	5.33 ± 0.65	12.15	106.60
	50	52.39 ± 1.04	1.99	104.79	53.48 ± 4.79	8.96	106.96
	250	270.27 ± 3.83	1.42	108.11	276.90 ± 18.04	6.52	110.76
5-Hydroxy omeprazole	5	4.53 ± 0.10	2.11	90.61	5.27 ± 0.59	11.12	105.32
	50	51.48 ± 2.96	5.76	102.96	52.29 ± 4.44	8.49	104.57
	250	274.62 ± 2.60	0.95	109.85	269.70 ± 21.81	8.09	107.88
Nifedipine	5	4.52 ± 0.31	6.89	90.38	4.43 ± 0.36	8.04	88.59
	50	50.39 ± 0.66	1.31	100.79	46.45 ± 5.28	11.37	92.89
	250	262.14 ± 22.27	8.49	104.85	254.12 ± 21.95	8.64	101.65
Dehydronifedipine	5	4.52 ± 0.40	8.94	90.32	4.62 ± 0.39	8.48	92.45
	50	48.02 ± 2.79	5.81	96.04	50.67 ± 1.28	2.52	101.33
	250	239.74 ± 11.09	4.63	95.90	272.04 ± 18.83	6.92	108.81
Dextromethorphan	5	4.50 ± 0.22	4.90	90.02	4.77 ± 0.33	6.91	95.46
	50	50.88 ± 1.25	2.47	101.76	49.75 ± 4.22	8.47	99.50
	250	269.65 ± 11.78	4.37	107.86	266.69 ± 20.50	7.69	106.68
Dextrorphan	5	4.43 ± 0.04	0.94	88.59	5.17 ± 0.54	10.36	103.43
	50	52.82 ± 5.04	9.53	105.65	52.13 ± 5.20	9.97	104.27
	250	276.68 ± 6.37	2.30	110.67	278.23 ± 28.29	10.17	111.29
Metformin	5	4.98 ± 0.19	3.82	99.59	5.25 ± 0.28	5.38	105.09
	50	51.78 ± 2.29	4.43	103.55	49.67 ± 5.39	10.85	99.33
	250	256.67 ± 11.28	4.39	102.67	235.72 ± 7.66	3.25	94.29
Furosemide	5	4.77 ± 0.46	9.55	95.43	4.59 ± 0.48	10.47	91.89
	50	45.98 ± 2.29	4.99	91.97	44.47 ± 2.55	5.74	88.95
	250	247.19 ± 9.61	3.89	98.88	242.15 ± 8.51	3.51	96.86
Valsartan	5	4.85 ± 0.45	9.23	97.03	4.87 ± 0.46	9.48	97.31
	50	49.07 ± 0.88	1.80	98.14	46.68 ± 4.90	10.51	93.35
	250	270.50 ± 5.55	2.05	108.20	249.08 ± 20.98	8.42	99.63
Digoxin	5	4.95 ± 0.06	1.20	99.00	4.98 ± 0.03	0.59	99.65
	50	44.16 ± 2.58	5.83	88.32	45.15 ± 2.73	6.06	90.29
	250	245.12 ± 13.38	5.46	98.05	248.52 ± 22.38	9.01	99.41
Methotrexate	5	5.00 ± 0.57	11.38	100.05	5.03 ± 0.57	11.39	100.59
	50	52.90 ± 4.62	8.74	105.80	45.11 ± 5.41	11.99	90.22
	250	270.17 ± 9.54	3.53	108.07	251.27 ± 21.74	8.65	100.51

The data expressed as the mean ± standard deviation (*n* = 6).

**Table 4 pharmaceutics-12-00938-t004:** Short-term, freeze-thaw cycle, and post-treatment stability of 15 probe substrates and metabolites.

Analyte	Spiked Concentration (ng/mL)	Short Term Stability (4 h, 25 °C)	Free-Thaw Cycle Stability (−80 °C/25 °C, 3 Cycles)	Post Treatment Stability (6 °C, 24 h)
Caffeine	5	96.9 ± 7.9	91.7 ± 10.7	92.4 ± 12.9
	250	105.3 ± 8.2	108.5 ± 7.7	109.2 ± 12.4
Paraxanthine	5	90.7 ± 9.8	80.6 ± 21.9	82.0 ± 15.7
	250	95.6 ± 3.9	107.8 ± 7.4	112.0 ± 10.5
Diclofenac	5	102.6 ± 10.5	108.3 ± 21.4	92.2 ± 14.6
	250	106.6 ± 10.7	102.3 ± 2.6	103.2 ± 3.3
4-Hydroxy diclofenac	5	98.1 ± 6.5	107.2 ± 26.2	91.6 ± 12.9
	250	100.4 ± 0.5	94.7 ± 5.6	108.8 ± 11.9
Omeprazole	5	101.3 ± 5.4	100.2 ± 17.8	96.5 ± 18.3
	250	105.0 ± 8.7	100.9 ± 4.9	109.4 ± 10.5
5-Hydroxy omeprazole	5	103.1 ± 3.1	101.8 ± 17.0	98.3 ± 7.4
	250	99.6 ± 2.8	104.6 ± 4.0	111.2 ± 9.8
Nifedipine	5	98.0 ± 1.8	94.4 ± 10.8	98.6 ± 7.3
	250	99.7 ± 1.9	109.5 ± 19.4	106.5 ± 16.3
Dehydronifedipine	5	96.2 ± 5.6	86.9 ± 12.5	91.6 ± 14.3
	250	103.1 ± 6.4	107.5 ± 14.9	111.8 ± 10.9
Dextromethorphan	5	94.8 ± 4.7	97.3 ± 9.0	91.1 ± 11.3
	250	101.7 ± 2.4	101.5 ± 4.0	114.5 ± 12.9
Dextrorphan	5	105.1 ± 5.7	98.9 ± 9.3	98.1 ± 10.2
	250	110.3 ± 9.7	96.6 ± 3.7	99.5 ± 4.5
Metformin	5	96.4 ± 18.2	104.8 ± 4.5	100.0 ± 3.7
	250	102.7 ± 6.8	106.5 ± 15.5	102.9 ± 14.2
Furosemide	5	98.4 ± 2.1	90.1 ± 9.2	89.7 ± 8.9
	250	101.3 ± 1.2	107.4 ± 18.4	105.5 ± 14.2
Valsartan	5	91.6 ± 14.3	101.5 ± 1.3	104.5 ± 5.1
	250	102.8 ± 3.5	103.9 ± 20.1	101.2 ± 12.9
Digoxin	5	97.9 ± 3.2	89.6 ± 9.2	99.1 ± 21.0
	250	96.9 ± 3.8	102.6 ± 16.4	97.8 ± 7.0
Methotrexate	5	96.6 ± 4.1	98.1 ± 19.4	92.6 ± 6.4
	250	102.7 ± 4.6	106.4 ± 21.3	109.7 ± 18.1

The data expressed as the mean ± standard deviation (*n* = 3).

**Table 5 pharmaceutics-12-00938-t005:** The comparison of pharmacokinetic parameters between single and CYP cocktail groups.

Cyps	Probe Substrate	Parameters	Single (*n* = 6)	CYP Cocktail (*n* = 6)
Cyp1a2	Caffeine	C_max_ (ng/mL)	866.9 ± 310.8	739.5 ± 216.8
		AUC_24 h_ (ng·h/mL)	3042.8 ± 1226.5	3965.1 ± 1632.2
		AUC_∞_ (ng·h/mL)	3051.3 ± 1231.0	3991.4 ± 1661.3
	Paraxanthine	AUC_24 h_ (ng·h/mL)	2878.6 ± 931.3	3593.8 ± 1119.0
		AUC_∞_ (ng·h/mL)	3135.4 ± 985.6	3152.4 ± 471.4
		MR	1.1 ± 0.4	1.0 ± 0.4
Cyp2c11	Diclofenac	C_max_ (ng/mL)	1863.9 ± 1309.8	1559.1 ± 1036.2
		AUC_24 h_ (ng·h/mL)	4386.5 ± 1264.3	5505.2 ± 1807.6
		AUC_∞_ (ng·h/mL)	4552.2 ± 1349.0	5741.9 ± 1836.8
	4-Hydroxy diclofenac	AUC_24 h_ (ng·h/mL)	4080.0 ± 2217.4	6944.7 ± 3761.8
		AUC_∞_ (ng·h/mL)	5831.5 ± 3529.0	7844.7 ± 4354.1
		MR	1.0 ± 0.6	1.2 ± 0.3
Cyp2c11	Omeprazole	C_max_ (ng/mL)	182.3 ± 158.5	62.0 ± 28.0
		AUC_24 h_ (ng·h/mL)	134.3 ± 107.6	145.1 ± 87.0
		AUC_∞_ (ng·h/mL)	134.5 ± 107.8	145.6 ± 87.0
	5-Hydroxy omeprazole	AUC_24 h_ (ng·h/mL)	116.6 ± 69.7	110.9 ± 63.8
		AUC_∞_ (ng·h/mL)	119.0 ± 71.4	112.0 ± 64.6
		MR	1.0 ± 0.2	0.8 ± 0.1
Cyp3a1	Nifedipine	C_max_ (ng/mL)	445.8 ± 202.0	420.1 ± 224.5
		AUC_24 h_ (ng·h/mL)	1392.2 ± 397.5	1490.9 ± 844.5
		AUC_∞_ (ng·h/mL)	1393.8 ± 397.0	1499.6 ± 841.9
	Dehydronifedipine	AUC_24 h_ (ng·h/mL)	22.3 ± 5.8	37.0 ± 15.8
		AUC_∞_ (ng·h/mL)	22.6 ± 5.8	37.4 ± 15.9
		MR	0.018 ± 0.01	0.028 ± 0.01
Cyp2d2	Dextromethorphan	C_max_ (ng/mL)	117.9 ± 78.1	180.0 ± 136.2
		AUC_24 h_ (ng·h/mL)	457.5 ± 373.3	971.1 ± 626.2
		AUC_∞_ (ng·h/mL)	459.1 ± 374.4	1011.1 ± 625.8
	Dextrorphan	AUC_24 h_ (ng·h/mL)	142.3 ± 137.0	196.9 ± 214.6
		AUC_∞_ (ng·h/mL)	208.6 ± 145.3	196.0 ± 121.9
		MR	0.30 ± 0.16	0.22 ± 0.20

C_max_: maximum concentration; AUC_24 h_ or AUC_∞_: area under the plasma concentration curve from zero to 24 h or infinite; MR: metabolic ratio calculated by dividing metabolite AUC_24 h_ by parent AUC_24 h_. The data expressed as the mean ± standard deviation.

**Table 6 pharmaceutics-12-00938-t006:** The comparison of pharmacokinetic parameters between single and transporter cocktail groups.

Transporters	Probe Substrate	Parameters	Single (*n* = 6)	Transporter Cocktail (*n* = 6)
Oct1/2	Metformin	AUC_24 h_ (ng·h/mL)	733 ± 402	767 ± 245
		AUC_∞_ (ng·h/mL)	750 ± 415	777 ± 247
		Xu (ng)	47,806 ± 29,071	52,474 ± 19,441
Oat1/3	Furosemide	AUC_24 h_ (ng·h/mL)	766 ± 370	652 ± 130
		AUC_∞_ (ng·h/mL)	773 ± 366	654 ± 130
		Xu (ng)	2330 ± 1285	6969 ± 3330
Oatp2	Valsartan	AUC_24 h_ (ng·h/mL)	2252 ± 247	2198 ± 518
		AUC_∞_ (ng·h/mL)	2285 ± 245	2225 ± 519
		Xu (ng)	920 ± 504	627 ± 441
P-gp	Digoxin	AUC_24 h_ (ng·h/mL)	4079 ± 716	3151 ± 1009
		AUC_∞_ (ng·h/mL)	4232 ± 828	3193 ± 1011
		Xu (ng)	29,954 ± 11,498	23,444 ± 10,659
Mrp2	Methotrexate	AUC_24 h_ (ng·h/mL)	1339 ± 242	1319 ± 252
		AUC_∞_ (ng·h/mL)	1362 ± 264	1329 ± 260
		Xu (ng)	2604 ± 1924	4621 ± 1431

AUC_24 h_ or AUC_∞_: area under the plasma concentration curve from zero to 24 h or infinite; Xu: excreted amount of substrate into urine for 24 h. The data expressed as the mean ± standard deviation.

**Table 7 pharmaceutics-12-00938-t007:** The comparison of pharmacokinetic parameters between the dual cocktail and CYP or transporter cocktail groups.

Cyps/Transporters	Probe Substrate	Parameters	CYP or Transporter Cocktail (*n* = 6)	Dual Cocktail (*n* = 7)
Cyp1a2	Caffeine	AUC_24 h_ (ng·h/mL)	3965 ± 1632	3533 ± 1788
		AUC_∞_ (ng·h/mL)	3991 ± 1661	3580 ± 1889
	Paraxanthine	AUC_24 h_ (ng·h/mL)	3593.83 ± 1119.05	2416.56 ± 1073.93
		AUC_∞_ (ng·h/mL)	3152.35 ± 471.42	2573.44 ± 1125.72
		MR	1.01 ± 0.37	0.77 ± 0.17
Cyp2c11	Diclofenac	AUC_24 h_ (ng·h/mL)	5505 ± 1808	5073 ± 1186
		AUC_∞_ (ng·h/mL)	5742 ± 1837	5116 ± 1200
	4-Hydroxy diclofenac	AUC_24 h_ (ng·h/mL)	6944.71 ± 3761.81	5184.02 ± 1609.43
		AUC_∞_ (ng·h/mL)	7844.69 ± 4354.05	5707.53 ± 2017.07
		MR	1.20 ± 0.34	1.02 ± 0.13
Cyp2c11	Omeprazole	AUC_24 h_ (ng·h/mL)	145 ± 87	191 ± 53
		AUC_∞_ (ng·h/mL)	146 ± 87	194 ± 52
	5-Hydroxyomeprazole	AUC_24 h_ (ng·h/mL)	110.93 ± 63.79	147.55 ± 68.57
		AUC_∞_ (ng·h/mL)	111.99 ± 64.62	150.42 ± 67.29
		MR	0.76 ± 0.08	0.74 ± 0.14
Cyp3a4	Nifedipine	AUC_24 h_ (ng·h/mL)	1491 ± 844	1320 ± 206
		AUC_∞_ (ng·h/mL)	1500 ± 842	1343 ± 204
	Dehydronifedipine	AUC_24 h_ (ng·h/mL)	37.03 ± 15.84	35.02 ± 10.94
		AUC_∞_ (ng·h/mL)	37.37 ± 15.90	36.99 ± 10.69
		MR	0.028 ± 0.012	0.025 ± 0.008
Cyp2d2	Dextromethorphan	AUC_24 h_ (ng·h/mL)	971 ± 626	654 ± 406
		AUC_∞_ (ng·h/mL)	1011 ± 626	674 ± 387
	Dextrorphan	AUC_24 h_ (ng·h/mL)	196.90 ± 214.62	155.63 ± 69.83
		AUC_∞_ (ng·h/mL)	196.02 ± 121.90	213.90 ± 74.20
		MR	0.20 ± 0.22	0.26 ± 0.08
Oct1/2	Metformin	AUC_24 h_ (ng·h/mL)	767 ± 245	1027 ± 349
		AUC_∞_ (ng·h/mL)	777 ± 247	1032 ± 348
		Xu (ng)	52,474 ± 19,441	48,885 ± 17,369
Oat1/3	Furosemide	AUC_24 h_ (ng·h/mL)	652 ± 130	701 ± 175
		AUC_∞_ (ng·h/mL)	654 ± 130	704 ± 174
		Xu (ng)	6969 ± 3330	4950 ± 2211
Oatp2	Valsartan	AUC_24 h_ (ng·h/mL)	2198 ± 518	2258 ± 674
		AUC_∞_ (ng·h/mL)	2225 ± 519	2289 ± 667
		Xu (ng)	627 ± 441	516 ± 338
P-gp	Digoxin	AUC_24 h_ (ng·h/mL)	3151 ± 1009	3302 ± 717
		AUC_∞_ (ng·h/mL)	3193 ± 1011	3430 ± 727
		Xu (ng)	23,444 ± 10,659	18,542 ± 7040
Mrp2	Methotrexate	AUC_24 h_ (ng·h/mL)	1319 ± 252	1432 ± 414
		AUC_∞_ (ng·h/mL)	1329 ± 260	1451 ± 421
		Xu (ng)	4621 ± 1431	5680 ± 3451

AUC_24 h_ or AUC_∞_: area under the plasma concentration curve from zero to 24 h or infinite; MR: metabolic ratio calculated by dividing metabolite AUC_24 h_ by parent AUC_24 h_; Xu: excreted amount of substrate into urine for 24 h. The data expressed as the mean ± standard deviation.

**Table 8 pharmaceutics-12-00938-t008:** The effect of ketoconazole and rifampin on the pharmacokinetic parameters of probe substrates of the dual cocktail and metabolites.

Cyps/Transporters	Probe Substrate	Parameters	Inhibition			Induction	
			Control (*n* = 6)	Ketoconazole (*n* = 4)	Rifampin (*n* = 4)	Control (*n* = 4)	Rifampin (*n* = 4)
Cyp1a2	Caffeine	AUC_24 h_ (ng·h/mL)	3104.8 ± 963.8	3579.8 ± 891.5	4131.9 ± 2129.2	3959.0 ± 1824.8	3687.9 ± 540.0
	Paraxanthine	AUC_24 h_ (ng·h/mL)	2416.6 ± 1073.9	3138.1 ± 1042.8	3555.2 ± 1740.2	3248.6 ± 819.6	3107.9 ± 351.6
		MR	0.8 ± 0.2	0.9 ± 0.1	0.9 ± 0.0	0.9 ± 0.4	0.9 ± 0.1
Cyp2c11	Diclofenac	AUC_24 h_ (ng·h/mL)	3551.7 ± 887.7	3863.3 ± 1109.8	3681.9 ± 724.2	4980.0 ± 1419.7	5744.9 ± 1647.1
	4-Hydroxy diclofenac	AUC_24 h_ (ng·h/mL)	3480.8 ± 1062.3	3584.7 ± 1222.2	3024.5 ± 459.7	5848.2 ± 2944.7	3986.8 ± 5627.2
		MR	1.0 ± 0.2	0.9 ± 0.2	0.8 ± 0.2	1.1 ± 0.3	0.6 ± 0.6
Cyp2c11	Omeprazole	AUC_24 h_ (ng·h/mL)	191.3 ± 53.3	403.5 ± 116.0 *	217.4 ± 18.3	207.9 ± 43.1	245.2 ± 31.2
	5-Hydroxyomeprazole	AUC_24 h_ (ng·h/mL)	119.0 ± 27.5	121.4 ± 37.1	105.0 ± 24.6	129.0 ± 17.8	133.4 ± 26.9
		MR	0.6 ± 0.1	0.3 ± 0.0 *	0.5 ± 0.1	0.6 ± 0.1	0.6 ± 0.1
Cyp3a1	Nifedipine	AUC_24 h_ (ng·h/mL)	1367.3 ± 201.1	5021.9 ± 3577.3 *	1597.3 ± 187.6	1595.2 ± 648.7	859.0 ± 268.2 *
	Dehydronifedipine	AUC_24 h_ (ng·h/mL)	60.2 ± 16.4	39.2 ± 11.1 *	67.5 ± 4.6	40.6 ± 14.7	56.0 ± 13.1
		MR	0.05 ± 0.02	0.01 ± 0.00 *	0.04 ± 0.00	0.0 ± 0.0	0.1 ± 0.0 *
Cyp2d2	Dextromethorphan	AUC_24 h_ (ng·h/mL)	1002.4 ± 206.0	1170.3 ± 228.8	1029.7 ± 455.1	1117.4 ± 574.1	970.1 ± 188.4
	Dextrorphan	AUC_24 h_ (ng·h/mL)	234.5 ± 167.5	142.5 ± 27.0	230.6 ± 43.1	229.2 ± 223.0	361.9 ± 205.8
		MR	0.2 ± 0.2	0.1 ± 0.0	0.2 ± 0.1	0.2 ± 0.2	0.4 ± 0.2
Oct1/2	Metformin	AUC_24 h_ (ng·h/mL)	908.0 ± 159.7	980.2 ± 90.3	788.1 ± 127.4	976.3 ± 406.4	1294.9 ± 468.0
Oat1/3	Furosemide	AUC_24 h_ (ng·h/mL)	700.9 ± 175.2	731.5 ± 123.6	682.6 ± 76.2	614.5 ± 83.9	524.2 ± 28.2
Oatp2	Valsartan	AUC_24 h_ (ng·h/mL)	2488.5 ± 315.6	2227.7 ± 1007.3	8693.9 ± 936.1 *	2258.4 ± 673.5	2333.4 ± 137.5
P-gp	Digoxin	AUC_24 h_ (ng·h/mL)	3302.0 ± 717.5	3677.2 ± 745.8	3732.4 ± 220.8	3302.0 ± 717.5	1349.5 ± 738.9 *
Mrp2	Methotrexate	AUC_24 h_ (ng·h/mL)	1432.1 ± 413.8	1229.0 ± 498.4	1332.9 ± 463.9	1432.1 ± 413.8	1294.0 ± 484.0

AUC_24 h_: area under the plasma concentration curve from zero to 24 h; MR: metabolic ratio calculated by dividing metabolite AUC_24 h_ by parent AUC_24 h_ * *p* < 0.05 statistically significant compared with control group by Mann-Whitney U test. The data expressed as the mean ± standard deviation.

**Table 9 pharmaceutics-12-00938-t009:** The composition and phenotypic index of dual cocktail.

Category	Isoforms in Rat (Human)	Probe Substrates (Dose, mg/kg)	Route of Administration	Phenotypic Index
CYPs	Cyp1a2 (CYP1A2)	Caffeine (1 mg/kg)	PO	AUC
	Cyp2c11 (CYP2C9)	Diclofenac (2 mg/kg)		AUC
	Cyp2c11 (CYP2C19)	Omeprazole (2 mg/kg)		AUC
	Cyp2d2 (CYP2D6)	Dextromethorphan (10 mg/kg)		AUC
	Cyp3a1 (CYP3A4)	Nifedipine (0.5 mg/kg)		AUC
Transporters	Oct1/2 (OCT1/2)	Metformin (0.5 mg/kg)	IV	AUC
	Oat1/3 (OAT1/3)	Furosemide (0.1 mg/kg)		AUC
	Oatp2 (OATP1B1/1B3)	Valsartan (0.2 mg/kg)		AUC
	P-gp (P-gp)	Digoxin (2 mg/kg)		AUC
	Mrp2 (MRP2)	Methotrexate (0.5 mg/kg)		AUC

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
