# Peer review of "The Development and Validation of a Novel “Dual Cocktail” Probe for Cytochrome P450s and Transporter Functions to Evaluate Pharmacokinetic Drug-Drug and Herb-Drug Interactions"

_pharmaceutics, 2020, doi:10.3390/pharmaceutics12100938_

Round 1

Reviewer 1 Report

The authors present an in vivo study regarding a cocktail method for simultaneous determination of CYP and transporter activities to evaluate potential pharmacokinetic drug interactions. The work is in the context of several other efforts that try to assess simultaneously potential PK interactions in vivo that could be mostly used in preclinical and clinical drug development or other studies such as drug-herb interaction studies (which this work also tries to apply). The validation steps followed were designed appropriately to evaluate any differences, the work is in this context interesting however there are several issues that has to be clarified prior to any considerations for publication. Generally, a major issue is the scientific impact today. Lines 416-417. In the era of pharmacometrics where simple in vitro studies can be scaled up to in vivo and/or clinical predictions with additional insights on the pharmacological mechanisms, what added value we gain from cocktail studies over single ones in which additional factors can be determined and controlled? For example, PK of metabolites and their potential impact cannot be determined in these studies since they may be overlapping. Moreover, what added value we gain from developing such protocols where we need approximately 150-200(?) animals only for the validation to perform acceptable statistical analysis.

Some additional comments follow:

1) Rats express CYP3A4 (and rest CYPS, transporters) or relative isoforms? Please edit or clarify.

2) Dextromethorpan is metabolized to 3-methoxymorphinan from CYP3A and then to 3-hydroxymorphinan from CYP2D6. Respectively dextrophan is a metabolite that derives from the activity of CYP2D6 and subsequently metabolized from CYP3A to hydroxymorphinan. Could this metabolic activity impact the results?

3) How did the authors ensure the avoidance of adverse drug reactions from the pharmacodynamic action of all those drugs? Is it safe for the animals to administer such large number of drugs  simultaneously? They designed a validation protocol to ensure avoidance of any PK-interactions but what was the extend of any potential PD interactions? 

4) Could these potential interactions impact the results (especially the PD ones)?

  • Diclofenac -methotrexate -> renal clearance (hematologic and GI toxicity)
  • Omeprazole - digoxin -> gastric pH (oral simultaneously)
  • Nifedipine - digoxin  -> MDR1 renal clearance
  • Metformin - digoxin -> renal tubular clearance
  • omeprazole - methotrexate -> renal clearance (PPI withdrawal with mtx)
  • furosemide - digoxin -> PD synergism
  • valsartan - diclofenac -> renal function deterioration
  • valsartan - digoxin -> decrease renal clearance
  • nifedipine - metformin -> PD antagonism
  • diclofenac - valsartan -> PD antagonism (increase serum potassium)
  • valsartan - furosemide -> decrease serum potassium
  • diclofenac - digoxin -> increase serum potassium
  • diclofenac - furosemide -> increase /decrease serum potassium
  • digoxin - furosemide -> increase / decrease serum potassium
  • methotrexate -digoxin -> decrease GI absorption (oral forms)
  • caffeine - methotrexate -> PD antagonism

5) Line 67 "Importantly, the probes were used at a lowest dose that can reduce a DDI possibility among cocktail probe substrates" What does it mean? Is there any DDI possibility (see previous comments 3 and 4)? On what basis the dosing was chosen? Any relative reference?

6) Interestingly not any reference to the relative guidance for industry from FDA is made. Why?

7) Why nifedipine for CYP3A and not any other substrate that is proposed from FDA for CYP3A mediated metabolism for drug interaction studies (sensitive or moderate sensitive substrate)

8) Please explain figure 1, line 124 and relative paragraph. The transporter cocktail was administered iv? How was assessed the impact on GI transporters? Rifampin PO would probably inhibit GI transporters (initially). Although Transporters cocktail are of low bioavailability, that would show big differences when the interaction study is conducted.

9) Ketoconazole is also inhibitor for P-gp. Could that impact the results? Not any relative comment made in manuscript

10) The LC-MS/MS method is novel and validated or based on previous ones? Considering line 222, if it is novel some additional data are required (even as supplementary, LLOQ, stability studies etc.).

11) Table 3. Nifedipine. Although not statistical significant -as the authors state- there is a double-fold difference in Cmax and wide variation that is not evident in figure 3 (even in log scale). Where is this difference attributed and why?

12) Figure 4. There is a huge variation in metformin and digoxin after 4 hrs. Please comment and explain the big variations after 4 hrs. Also metformin Xu is half (table 4)

13) What ketoconozaole has to do with omeprazole CYP2C19? Ketoconazole may inhibit CYP2C19 in vitro but in vivo is a weak inhibitor. How that interaction occurred? Generally, PPIs may decrease the gastrointestinal absorption of the azole antifungal agents (such as azoles, ketoconazole) which require acidic environment for dissolution. Moreover there are some evidence regarding potential impact on the formation of omeprazole sulfone from ketoconazole which however is mediated from CYP3A not CYP2C19 which catalyzes omeprazole hydroxylation.

14) Under what background evidence the RGE study was conducted? Any prior knowledge for potential interaction or just a demonstration example? Apart of how the administration was made there is poor description about RGE. What red ginseng extract, how the standardization of compounds was conducted? How the authors ensured similar content of herb administration among the animals in the vivo study? There is only the description of the same protocol administration and the presentation of the results.

15) Major English edit is required throughout text.

Reviewer 2 Report

The manuscript describes the development of a dual cocktail probe to evaluate pharmacokinetic interactions with regards to transporter and metabolic enzyme modulation.  The work is well executed and presented and of high interest to readers.  The following minor issues should be addressed:

1)Section 2.1: Please provide full details of the red ginseng extract that was used in the study as well as phytochemical characterisation thereof.

2) Section 2.3, line 172: Please provide the rationale for choosing a concentration of 1.4 g/kg of red ginseng extract with a citation to where this dose was recommended.

3) Please correct the following English grammar errors: line 39:  the word potential is used in duplicate - please remove second "potential"; line 115: "an" should be replaced with "the"; line 133: replace "from in" with "within"; line 219: replace "were" with "are".

Round 2

Reviewer 1 Report

The authors addressed successfully the initial comments regarding their manuscript. Overall their scientific work is adequately presented. My only concern is that although it is a well developed and approached study, the IV administration of the transporter cocktails subtracts the capability to sufficiently study GI-DDIs or GI-HDIs. They also comment it n line 549. The work is interesting in the context of developing a cocktail method for PK DDIs (or HDIs). The manuscript could be further processed for publication.